# Enhancing Entomological Surveillance: Real-Time Monitoring of Mosquito Activity with the VECTRACK System in Rural and Urban Areas

**DOI:** 10.3390/biology14081047

**Published:** 2025-08-14

**Authors:** Manuel Silva, Bruna R. Gouveia, José Maurício Santos, Nélia Guerreiro, Alexandra Monteiro, Soraia Almeida, Hugo Costa Osório

**Affiliations:** 1National Institute of Health Doutor Ricardo Jorge (INSA), Centre for Vectors and Infectious Diseases Research (CEVDI), Avenida da Liberdade n.-5, 2965-575 Águas de Moura, Portugal; 2Direção Regional da Saúde, Rua 31 de Janeiro, n°. 54 e 55, 9054-511 Funchal, Portugal; bruna.gouveia@madeira.gov.pt (B.R.G.); mauricio.f.santos@madeira.gov.pt (J.M.S.); 3Interactive Technologies Institute—LARSyS, Polo Científico e Tecnológico da Madeira, Caminho da Penteada, Piso-2, 9020-105 Funchal, Portugal; 4Departamento de Saúde Publica e Planeamento, Administração Regional de Saúde do Algarve, IP, EN 125, Sítio das Figuras, Lote 1, 2° andar, 8005-145 Faro, Portugal; nguerreiro@dgs.min-saude.pt (N.G.); amonteiro@dgs.min-saude.pt (A.M.); 5Unidade de Saúde Pública, Administração Regional de Saúde do Algarve, IP, EN 125, Sítio das Figuras, Lote 1, 2° andar, 8005-145 Faro, Portugal; sfalmeida@ulsalg.min-saude.pt; 6Instituto de Saúde Ambiental (ISAMB) Environment and Infectious Diseases Research Group, Av. Prof. Egas Moniz, Ed. Egas Moniz, Piso 0, Ala C, 1649-028 Lisboa, Portugal

**Keywords:** mosquito surveillance, VECTRACK, *Aedes* vectors, machine-learning

## Abstract

*Aedes* mosquitoes are primary vectors of arboviruses such as dengue, Zika, and chikungunya, presenting substantial public health concerns. In Portugal, *Aedes aegypti* was first detected in Madeira in 2005, and *Aedes albopictus* emerged in mainland regions by 2017. These invasions underscore the need for effective entomological surveillance, which is traditionally labor-intensive. This study evaluates the field performance of VECTRACK, a bioacoustic sensor system integrated with Biogents Sentinel traps, designed for automated mosquito identification. VECTRACK units were deployed across three Portuguese regions—Funchal (Madeira), Palmela, and the Algarve. Mosquitoes were manually collected and identified, and results were compared with automated sensor outputs using Spearman’s rank correlation. A total of 176 mosquitoes were captured in Madeira, 732 in Palmela, and 143 in the Algarve. Strong correlations were observed between manual and sensor identifications (Spearman’s ρ for females and males in Madeira: 0.84 and 0.92; Palmela: 0.99 and 0.84; Algarve: 0.98 and 0.99; all *p* < 0.001), confirming the sensor’s reliability. The VECTRACK system demonstrated the ability to accurately distinguish mosquitoes from non-target insects, differentiate between *Aedes* from *Culex* genera, and identify the sex of captured specimens. These findings support its potential for scalable, real-time vector surveillance and early outbreak detection, particularly in areas facing increasing risk of arboviral transmission.

## 1. Introduction

Mosquitoes of the genus *Aedes* are capable of transmitting multiple viruses, including dengue, Zika, and chikungunya [1]. Notably, the species *Aedes albopictus*, commonly known as the Asian tiger mosquito, has been increasing its distribution across Europe and is now found in countries such as France, Italy, Spain, and Portugal [2,3,4,5,6]. In Portugal, it was first detected in the northern region in 2017 [3], and its distribution has been increasing since then [7]. Its initial establishment in Portugal was likely facilitated by the passive transport of the eggs via goods and adults inside vehicles, combined with a climatic context conducive to its establishment [8].

Another important invasive mosquito species, *Aedes aegypti*, can also be found in parts of Europe, including Madeira Island (Portugal) and Cyprus [9]. *Aedes aegypti* was first detected on Madeira in 2005 and has since shown significant expansion in its geographical distribution, particularly colonizing densely populated areas along the island’s southern coast [10,11]. In 2012, this species was responsible for the first autochthonous dengue fever outbreak in Europe since 1920, resulting in 1080 laboratory-confirmed cases and over 2000 probable infections [12,13].

Climatic projections indicate that both species could further expand their geographical range, particularly as global warming intensifies and winters become milder [14]. These findings underscore the urgent need for coordinated national and regional strategies focused on enhanced mosquito vector surveillance, public education, and sustainable control measures. Additionally, studying the behavioral ecology and genetic characteristics of these invasive populations is essential to develop targeted interventions to mitigate their impact on public health.

As the geographic distribution of mosquito vectors continues to expand, mosquito monitoring systems and programs must evolve to become more efficient, faster, and autonomous. While monitoring immature stages can be cost-effective, requiring only ovitraps and larval surveys, it is not suitable for estimating adult mosquito abundance due to the weak correlation between egg, larval, and pupal indices and adult population levels [15]. Moreover, studies focusing on seasonal variation in mosquito abundance have shown that adult trap monitoring provides more reliable data than indices based on immature stages [16]. Consequently, adult mosquito surveillance is a critical key element for risk assessment and vector control strategies, particularly in contexts where accurate estimates of adult abundance are essential for guiding targeted interventions [17].

Most adult mosquito traps rely on attractants, such as chemical lures or carbon dioxide (CO_2_), and use a fan to draw approaching insects into a catch bag [18,19]. However, these traps require regular human intervention to periodically inspect the catch bag and identify the collected samples. The delay between manual inspections can range from several days to a week, which can introduce discrepancies between what enters the trap and what is ultimately identified. These discrepancies may result from factors such as predation, decomposition, or physical damage to the specimens, potentially compromising data accuracy.

To address this issue, optical methods, such as technologies that detect insects by measuring how their movement modulates light beams, typically using infrared or laser sensors to capture flight patterns and wingbeat frequencies, have been employed to study the interaction between insect flight and light [20,21,22]. Although a variety of sensors incorporating artificial intelligence have been developed over the years, their accuracy has varied depending on the specific approach. Nonetheless, the findings suggest that such technologies offer a viable alternative to manual identification methods [23,24]. This highlights the need for innovative approaches to traditional surveillance. One such approach involves the use of “smartraps”, automated trapping devices equipped with sensors and embedded intelligence (e.g., software or machine learning algorithms) capable of classifying captured insects in real time. These devices generate digital records and telemetry data without requiring manual specimen sorting, enabling the automatic and remote classification of mosquitoes with minimal time delay. Smartraps typically employ optical sensors to detect multiple biological features of mosquitoes as they enter the trap, such as wingbeat and mosquito size. This is followed by analysis methods utilizing machine learning algorithms to identify the mosquitoes in real time [25,26,27,28,29]. The newly developed bioacoustic sensor, henceforth referred to as the VECTRACK system, was developed by IRIDEON (SL. Barcelona, Spain) and tested by the Centre for Vectors and Infectious Diseases Research, National Institute of Health (CEVDI/INSA), within the framework of the Portuguese National Vector Surveillance Network (REVIVE). The first deployment took place in Faro, Algarve, in a peri-urban setting where *Aedes albopictus* is established. This setting provided a valuable opportunity to evaluate the system’s performance in detecting and classifying invasive mosquito species under real field conditions. The second deployment was conducted in Funchal, Madeira Island, an urban environment where *Aedes aegypti* has been established for nearly two decades. The third deployment was conducted in Palmela, mainland Portugal, in a rural area characterized by high mosquito species diversity but no known presence of invasive mosquitoes.

This study aims to validate the accuracy and field performance of the VECTRACK system for real-time mosquito surveillance across rural, peri-urban, and urban ecological contexts. We assess the system’s capability to identify mosquito genus and sex, as well as its effectiveness in distinguishing target species from non-target arthropods under field conditions.

## 2. Materials and Methods

### 2.1. VECTRACK System

The VECTRACK sensor incorporates a pair of optoelectronic arrays—comprising a light-emitting source and a corresponding photodetector panel—aligned across a transparent cylindrical passage with a diameter of 105 mm. The emitter consists of a 2D matrix of infrared LEDs (940 nm), while the receiver comprises a matching array of photodiodes. The active sensing path spans 70 mm. Signal output is amplified and digitized via an analogue-to-digital converter at a sampling frequency of 9603 Hz. Detection of a mosquito within the sensing region initiates automatic data acquisition, capturing up to 1024 samples over a maximum interval of 107 milliseconds. Each detection event is time-stamped, and ambient temperature is recorded concurrently [29]. The sensor was trained using a database comprised exclusively of *Aedes albopictus* and *Culex pipiens* specimens reared under controlled insectary conditions.

The sensor was attached to the inlet of a commercially available BG-Mosquitaire suction trap (Biogents AG, Regensburg, Germany) for field-based vector monitoring. Designed to attract primarily host-seeking female mosquitoes, the trap utilizes a combination of visual and chemical cues and performs effectively across different mosquito densities. Solid carbon dioxide (dry ice) was deployed as a CO_2_ source above the entrance.

### 2.2. Site Selection

Sampling site selection was guided by the target mosquito species: *Aedes albopictus* in the Algarve region, *Aedes aegypti* on Madeira Island, and other *Aedes* species in Palmela. All mosquito collection sites had previously been surveyed under the REVIVE project [29,30,31,32], providing valuable baseline data on local mosquito diversity and abundance.

Based on these criteria, the VECTRACK system was deployed at (1) a peri-urban plant nursery area in the municipality of Loulé, Algarve (coordinates: 37.10134887, −8.12356948); (2) a small urban enclave near a main street in Funchal, Madeira Island (coordinates: 32.651318, −16.908259); and (3) a rural site in the municipality of Palmela (coordinates: 38.583248, −8.689730).

### 2.3. Periodicity of Inspections

In the Algarve, inspections were conducted at intervals ranging from 7 to 17 days, with the catch bag replaced between visits. Collections took place from 24 June to 17 September 2021. A total of seven catch bags were retrieved, and all captured insects were identified and counted.

In Palmela, the VECTRACK system was inspected daily on weekdays between 10:00 and 11:00 AM, from 8 June to 23 June 2021. Over this period, ten catch bags were collected, and the insects were subsequently identified and counted.

In Funchal, the VECTRACK system operated weekly, being activated on Monday mornings and deactivated on Friday afternoons, with the catch bag collected immediately afterward. This surveillance period extended from 8 August 2022 to 28 June 2023. A total of 38 catch bags were analyzed, and the insects were identified and counted.

### 2.4. Mosquito Identification and Data Analysis

Upon arrival at the laboratory, captured insects were manually identified by an entomologist under a stereo microscope (SZX7, Olympus, Tokyo, Japan). These identifications were then compared to those generated by the VECTRACK system via the SENSCAPE application platform (https://senscape.eu/login, accessed on 7 July 2025), which is accessible through any standard internet browser. The platform displays VECTRACK identifications in various formats, including histograms, individual sample timelines, and daily sample counts.

Two operational capabilities of the VECTRACK system were assessed: (i) its accuracy in distinguishing target mosquito genera (*Aedes* and *Culex*) from non-target arthropods, and (ii) its precision in identifying the genus and sex of captured specimens. To evaluate the agreement between automated sensor detections and manual identifications, both correlation and linear regression analyses were performed. Temporal and quantitative concordance between the two datasets was visualized using time-series plots and scatter diagrams for each sampling interval.

The Pearson correlation coefficient (r) and its associated *p*-value were used to quantify the strength and statistical significance of the relationship between sensor-based and manual insect counts. Linear regression analysis yielded the slope, intercept, and coefficient of determination (R^2^), collectively describing how closely the sensor estimates aligned with manually obtained data. A slope significantly different from unity indicated systematic bias in the sensor’s output, suggesting either consistent overestimation or underestimation.

To assess the classification performance of the VECTRACK, confusion matrices were constructed for four distinct scenarios: (i) discrimination between target and non-target insects, (ii) genus-level classification (*Aedes* or *Culex*), and sex-specific classification for (iii) *Aedes* and (iv) *Culex*. Manual identifications served as the reference standard. For each scenario, the number of true positives (TPs), false positives (FPs), true negatives (TNs), and false negatives (FNs) was determined. The positive class was defined contextually, and TP was taken as the minimum value between sensor and manual counts. Discrepancies were categorized as FP or FN, depending on whether the sensor overcounted or undercounted, respectively. TN values were derived by subtracting FP from the total number of manually identified negative cases.

Based on the confusion matrix values, standard performance metrics were calculated to quantitatively assess the sensor’s classification reliability under field conditions. These included classification accuracy [(TP + TN)/total observations], sensitivity [TP/(TP + FN)], specificity [TN/(TN + FP)], and precision [TP/(TP + FP)]. Together, these metrics provided a quantitative evaluation of the sensor’s ability to correctly identify and classify mosquito specimens in real-time field deployment.

## 3. Results

### 3.1. Manual Identifications in the Sampling Site Algarve

A total of 143 mosquitoes representing four genera—*Aedes*, *Anopheles*, *Culiseta*, and *Culex*—were collected in the Algarve region, out of which 132 individuals were identified as belonging to the target species group. Within the target group, the distribution by genus and sex, in descending order of relative abundance, was as follows: *Culex pipiens* females (79.0%), *Aedes albopictus* females (10.5%), and *Culex pipiens* males (2.8%) (Table 1). Additionally, a table containing the captures of all three sampling sites in available in Appendix A. During the same sampling period, a total of 902 non-culicid insects—such as chironomids, moths, and fungus gnats—were also captured.

### 3.2. VECTRACK Identifications in the Sampling Site Algarve

In total, the VECTRACK system identified 161 target mosquitoes, demonstrating a strong correlation with manual identification (r = 0.987, *p*-value = 0.999). This high level of agreement is shown in Figure 1 and remains consistent when the comparison is based on the date of capture (Figure 2).

Concerning the classification of total mosquitoes, the coefficient of determination (R^2^) was 0.98 (Figure 3). When analyzing the combined dataset, target genera (*Culex* and *Aedes*), and sexes, a final R^2^ value of 0.90 was obtained (Figure 4), further supporting the overall effectiveness of the VECTRACK system in mosquito identification under field conditions.

Table 2 demonstrates that the VECTRACK system consistently achieved high performance across all evaluated metrics in distinguishing target from non-target insects. The system achieves comparable accuracy in classifying specimens between the *Aedes* and *Culex* genera, as well as in differentiating sexes within the *Culex* genus. In contrast, the classification accuracy for male and female *Aedes* specimens was lower, likely due to the limited number of individuals from this group collected during sampling.

### 3.3. Manual Identifications in the Sampling Site Palmela

A total of 732 mosquitoes were captured in Palmela, of which 713 belonged to the target species across four genera: *Aedes*, *Anopheles*, *Culiseta,* and *Culex*. Among the target mosquitoes, the distribution by genus and sex from most to least abundant was as follows: *Culex theileri* female (83.1%), *Culex theileri* male (4.2%), *Aedes caspius* female (3.9%), *Culex pipiens* female (3.6%), *Culex pipiens* male (0.9%), and *Aedes caspius* male (0.1%) (Table 3). In addition, 885 non-mosquito insects were collected during the same sampling period.

### 3.4. VECTRACK Identifications in the Sampling Site Palmela

The VECTRACK system identified 730 target mosquitoes, demonstrating a strong correlation with the manual identification method (Pearson’s r = 0.987 and *p*-value = 0.999). This high level of agreement is illustrated in Figure 5 and remains consistent when the comparison is stratified by date of capture (Figure 6).

In all classification scenarios, the coefficient of determination (R^2^) exceeded 0.95 (Figure 7), indicating a strong agreement between manual and sensor-based identifications. The only exception was observed for male *Aedes* spp., which yielded an R^2^ value of 0.071, most likely due to the very low number of specimens in this category. When considering the combined dataset, target genera (*Culex* and *Aedes*), and both sexes, a final R^2^ value of 0.975 was achieved (Figure 8), further supporting the overall reliability of the VECTRACK system in mosquito identification.

As demonstrated in Table 4, the VECTRACK system consistently achieved high performance across all metrics for distinguishing target from non-target species. Similarly strong results were observed in the classification between the *Aedes* and *Culex* genera, as well as in sex differentiation within the *Culex* genus. In contrast, classification accuracy for male versus female *Aedes* individuals was considerably lower, likely due to the limited number of specimens collected for this group during the sampling period.

### 3.5. Manual Identifications in the Sampling Site of Funchal

In total, 176 mosquitoes were collected between August 2021 and April 2022 (Table 5). Two species were identified: *Aedes aegypti* (n = 160), comprising 83 males, 65 females, and 12 specimens whose sex could not be determined; and *Culex Pipiens* (n = 16), including 1 male, 13 females, and 2 unclassified by sex. During the same sampling period, 191 non-mosquito insects were also captured.

### 3.6. VECTRACK Identifications in the Sampling Site Funchal

The VECTRACK system identified a total of 171 mosquitoes, including 145 Aedes aegypti (100 males and 45 females) and 26 *Cx. pipiens* (8 males and 18 females). This total was five mosquitoes fewer than those recorded through manual inspection. Despite this discrepancy, the system demonstrated a strong correlation with manual counts (Pearson’s r = 0.987 and *p*-value = 0.999). This high level of agreement is illustrated in Figure 9 and remains consistent when the comparison is stratified by the date of capture (Figure 10).

In line with the Palmela trial, most classification scenarios achieved a coefficient of determination (R^2^) above 0.75, indicating strong agreement between automated and manual identifications (Figure 11). When considering the combined dataset, target genera (*Aedes* and *Culex*), and both sexes being aggregated, the system achieved an overall R^2^ of 0.7499 (Figure 12), indicating strong general performance of the VECTRACK system in mosquito identification tasks under diverse field conditions.

As shown in the summary of the confusion matrix (Table 6), the system shows high percentages across all parameters regarding the separation between target and non-target species. These promising results continue in both the separation between the *Culex* and *Aedes* genera and the differentiation of the sexes in the *Aedes* genus. However, the results for the differentiation of male and female *Culex* are much lower than the other, possibly due to the reduced presence of this genus in the captures.

## 4. Discussion

This study evaluated the performance of the VECTRACK automated mosquito classification system under field conditions. The system integrates a commercial mosquito suction trap, an optical sensor, and a machine learning pipeline, capable of distinguishing target mosquito genera (*Aedes* and *Culex*) from non-target insects entering the trap. Additionally, it classifies the genus and sex of the target mosquitoes. The system provides high-resolution temporal data on mosquito population dynamics, with time stamps accurate to one second [28]. The machine learning model was trained using recordings from thousands of laboratory-reared mosquitoes, which flew through the sensor under varying temperature and age conditions. Across all field trials, the system achieved an average balanced accuracy of 98% in differentiating target mosquitoes from other flying insects, indicating a high level of reliability in mosquito detection and classification.

The number of replicates varied across study sites—7 in Algarve, 38 in Funchal, and 10 in Palmela—reflecting practical differences such as site accessibility, monitoring duration, and unpredictable field conditions commonly encountered in entomological surveys. The larger sample size in Funchal enabled more detailed temporal analyses and provided greater statistical power for detecting population trends. In contrast, the lower number of replicates in Algarve and Palmela settings was influenced by local constraints and logistical limitations inherent to field-based vector surveillance. To ensure the robustness of data interpretation despite these disparities, all statistical analyses, including correlation assessments and regression models, were conducted independently for each site. This approach minimized the risk of bias due to unequal sample sizes. Rather than directly comparing results across locations, it highlighted that unequal sampling frequency may limit the detection of rare or short-lived fluctuations in mosquito abundance, particularly at sites with fewer collections. Nevertheless, the consistently high agreement between manual identifications and automated sensor results across all three sites demonstrates the VECTRACK system’s reliability, even under varying operational conditions.

In line with our results, the VECTRACK system offers several advantages over traditional manual mosquito surveillance methods: (i) it substantially reduces the manual effort required to process catch bags, particularly when sorting target mosquitoes from a large number of non-target insects, and recording results; (ii) it delivers classification results significantly faster than conventional monitoring programs, which typically operate on 7- to 15-day collection cycles, thereby enabling quicker epidemiological responses when needed; (iii) it is unaffected by sample degradation or predation, preserving data integrity; (iv) it assigns a precise timestamp to each capture event, allowing for the tracking of mosquito activity patterns with temporal resolutions as fine as one second; and (v) it automatically generates tabular and graphical outputs, which can be downloaded or integrated into risk-mapping platforms through an application programming interface (API).

Compared to other automated mosquito classification systems, the VECTRACK system offers additional advantages: (i) it performs in-field classification of both genus and sex for target mosquitoes, which has not been previously reported in the scientific literature; (ii) it achieves high classification accuracy (above 85% for all three trials) for *Aedes* and *Culex* mosquito genera across a range of temperatures in which these mosquitoes are typically active, regardless of their relative abundance in the catch bag; (iii) it is compatible with commercially available mosquito traps commonly used in routine entomological surveillance, allowing for manual inspection when necessary [22,24,26].

In addition to the automated classification capabilities, the VECTRACK system is integrated with the SENSCAPE platform, which provides real-time access to surveillance data through both a web-based dashboard and a mobile application. The mobile app is currently available on the Google Play Store for Android devices, allowing field technicians and public health officials to monitor mosquito activity remotely and promptly respond to emerging risks. This functionality greatly enhances the operational value of the system by minimizing the need for on-site presence while ensuring continuous situational awareness.

Despite the robust performance of the VECTRACK system in classifying mosquitoes by genus and sex, some limitations should be noted. The system exhibited lower accuracy for underrepresented groups, particularly male *Aedes* spp., which were not frequently captured during field sampling. This class imbalance likely restricted model sensitivity, as also observed in other studies using machine learning tools for entomological surveillance. Additionally, high densities of insects within the trap can cause overlapping flight signals, potentially leading to misclassification due to interference in optical detection.

The current sensor was trained exclusively with laboratory-reared *Aedes albopictus* and *Culex pipiens*, which may constrain its applicability to other mosquito species or different ecological regions. While the system showed high concordance with manual identification across three Portuguese regions (Algarve, Palmela, and Funchal), unequal sample sizes at these sites may have reduced the detection of rare or rapid population shifts—emphasizing the need for further validation in broader ecological settings and expanded training datasets for general field use.

This technology has proven to be a valuable tool for monitoring both species of *Aedes* and *Cx. pipiens* populations in urban and peri-urban settings, two of the primary vector species in temperate areas, and shows further potential [29]. However, additional field trials, particularly in rural areas and with other mosquito species, including indigenous *Aedes* and *Culex* species, are necessary to fully validate these findings. More extensive training with additional biological traits, such as different developmental stages of the female’s gonotrophic cycle, would also be beneficial. Additionally, genus identification may be insufficient in some regions to gather information about vectorial capacity for arboviruses. Therefore, to expand the system’s applicability across a broader range of geographical regions, it would be desirable to achieve a species-level identification by training the sensor with other medically important vector species such as *Aedes aegypti*, *Culex quinquefasciatus*, and the *Anopheles gambiae* complex.

## 5. Conclusions

This study demonstrated the effectiveness of a novel bioacoustic sensor system, VECTRACK, in identifying the genus and sex of *Aedes* and *Culex* mosquitoes when integrated with a commercial suction trap in both field conditions. These results support the system’s applicability for vector surveillance and highlight its potential to significantly reduce both the response time and workload associated with traditional mosquito monitoring by national health authorities. Furthermore, the integration of this novel bioacoustic sensor into existing surveillance programs in areas where these vector species are present could significantly improve the quality of the results. By enabling automated, high-resolution, and real-time mosquito classification, the VECTRACK system represents a valuable tool for strengthening entomological surveillance and improving public health preparedness.

## Figures and Tables

**Figure 1 biology-14-01047-f001:**
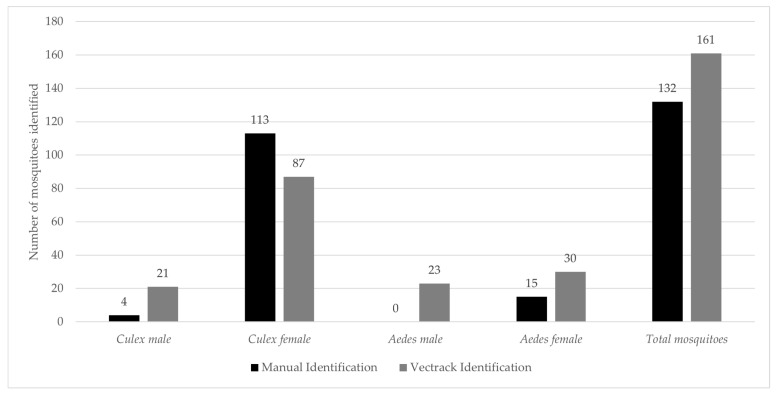
Comparison between the total number of target mosquitoes manually identified and those classified by the VECTRACK system in the Algarve region (coordinates: 37.10134887, −8.12356948), by genus and sex. The study was conducted between 24 June and 17 September 2021.

**Figure 2 biology-14-01047-f002:**
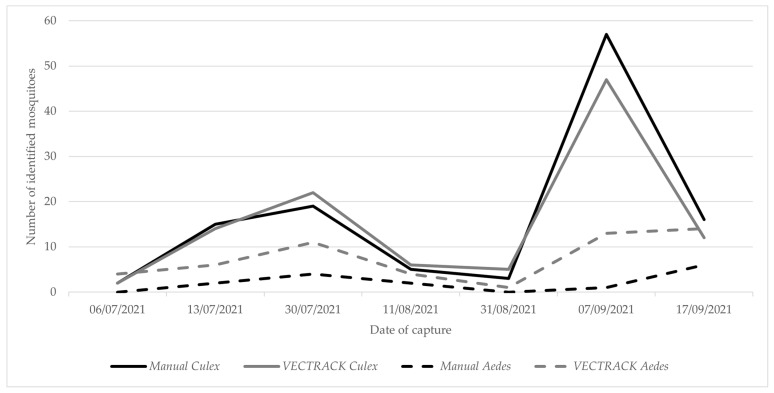
Comparison of the number of target mosquitoes manually identified and those classified by the VECTRACK system, by date of capture in the Algarve region (coordinates: 37.10134887, −8.12356948), by genus and date of capture. The study was conducted between 24 June and 17 September 2021.

**Figure 3 biology-14-01047-f003:**
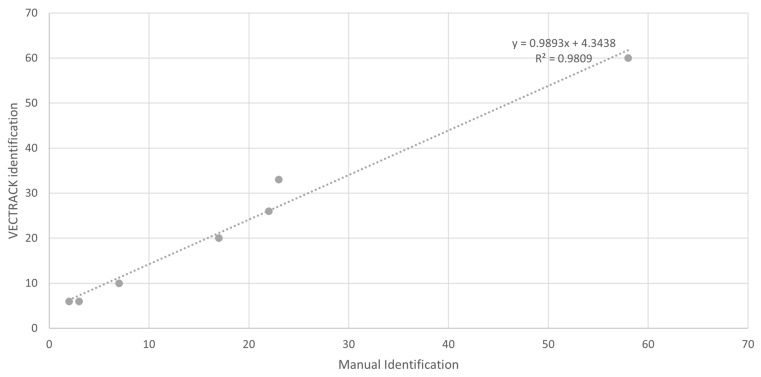
Scatter plot and linear regression analyses of VECTRACK sensor count versus manual mosquito counts in Algarve (coordinates: 37.10134887, −8.12356948), based on total mosquito captures recorded between 8 June and 23 June 2021.

**Figure 4 biology-14-01047-f004:**
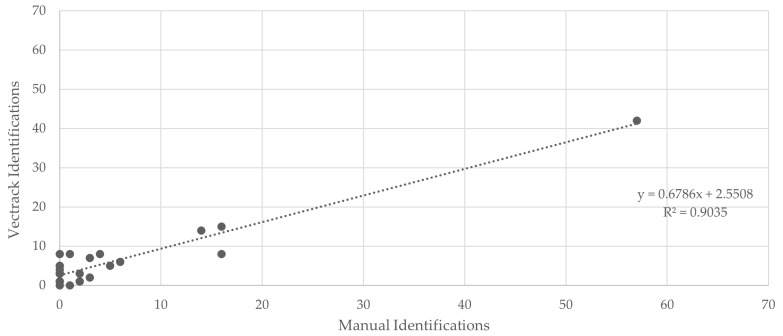
Scatter plot and linear regression analysis comparing VECTRACK sensor identifications with manual identifications in Algarve (coordinates: 37.10134887, −8.12356948), considering both mosquito genus and sex, based on total mosquito captures. Recorded between 24 June and 17 September 2021.

**Figure 5 biology-14-01047-f005:**
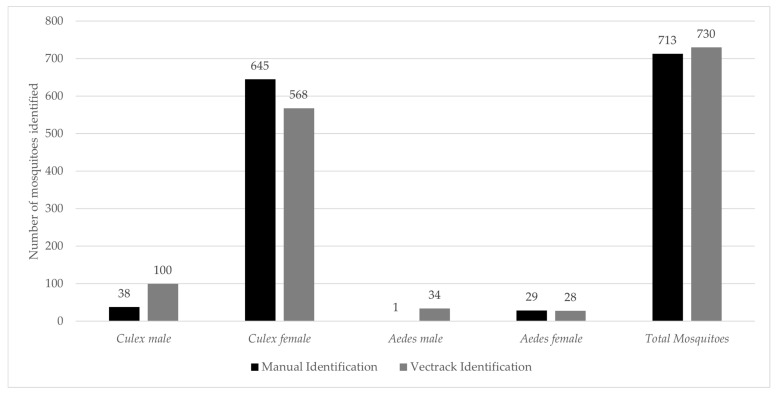
Comparison of the total number of target mosquitoes captured and manually identified with those identified by the VECTRACK system in Palmela (coordinates: 38.583248, −8.689730), categorized by genus and sex. Both the captures and VECTRACK identifications were conducted between 8 June and 23 June 2021.

**Figure 6 biology-14-01047-f006:**
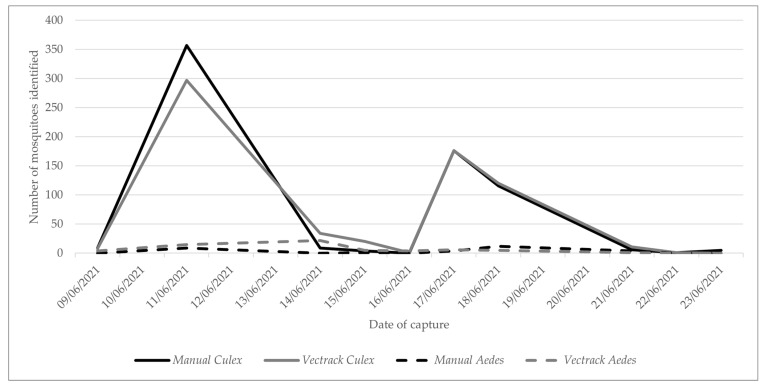
Comparison of the number of target mosquitoes captured and manually identified with those identified by the VECTRACK system in Palmela (coordinates: 38.583248, −8.689730), by genus and date of capture. Both captures of VECTRACK identification were conducted between 8 June and 23 June 2021.

**Figure 7 biology-14-01047-f007:**
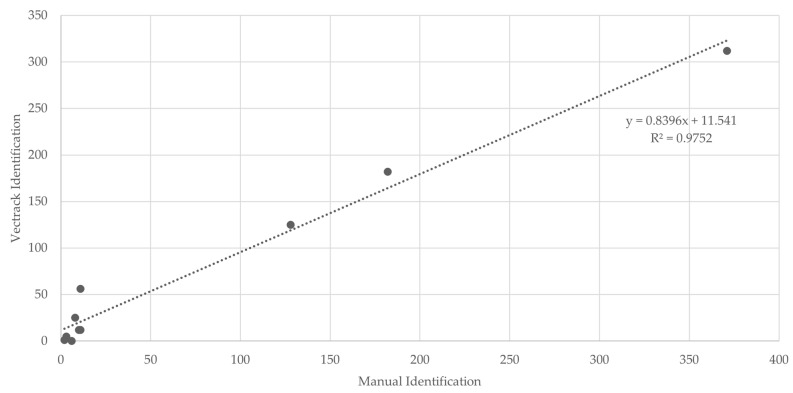
Scatter plot and linear regression analyses of VECTRACK sensor count versus manual mosquito counts in Palmela (coordinates: 38.583248, −8.689730), based on total mosquito captures. Recorded between 8 June and 23 June 2021.

**Figure 8 biology-14-01047-f008:**
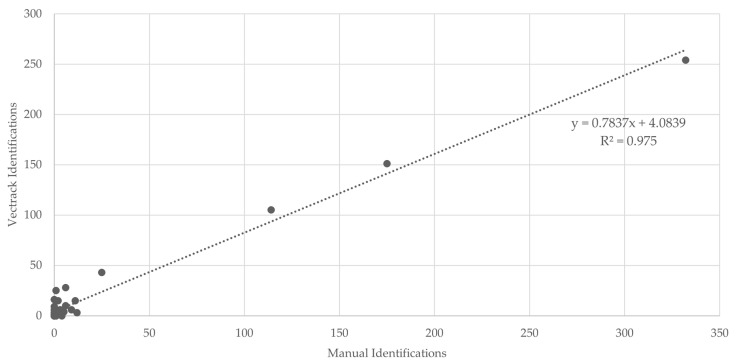
Scatter plot and linear regression analyses of VECTRACK sensor count versus manual mosquito counts in Palmela (coordinates: 38.583248, −8.689730), considering both mosquito genus and sex, based on total mosquito captures. Recorded between 8 June and 23 June 2021.

**Figure 9 biology-14-01047-f009:**
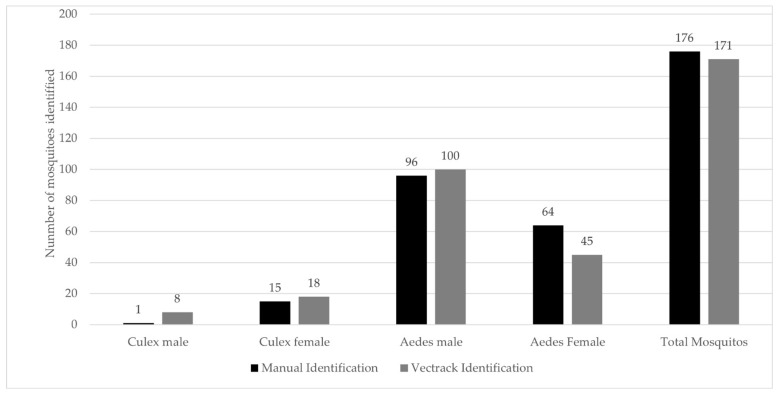
Comparison between the total number of target mosquitoes manually identified and those classified by the VECTRACK system in Funchal (coordinates: 32.651318, −16.908259) by genus and sex. The study was conducted between 8 August and 28 June 2023.

**Figure 10 biology-14-01047-f010:**
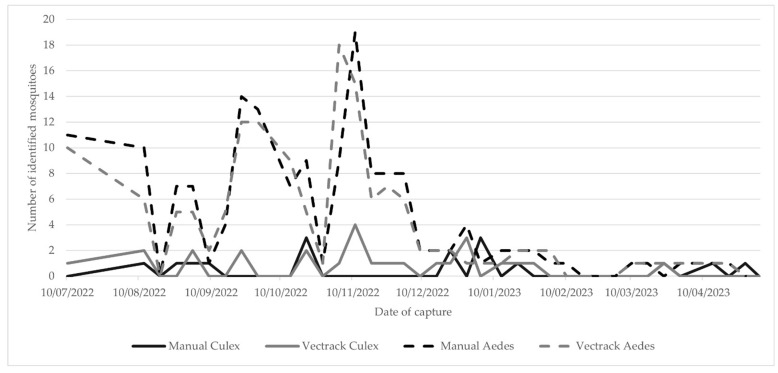
Comparison of the number of target mosquitoes manually identified and those classified by the VECTRACK system, by date of capture in Funchal (coordinates: 32.651318, −16.908259), and by genus and date of capture. The study was conducted between 8 August and 28 June 2023.

**Figure 11 biology-14-01047-f011:**
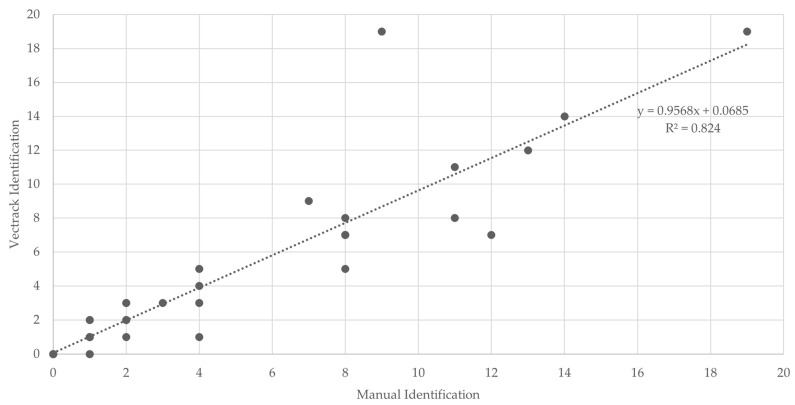
Scatter plot and linear regression analyses of VECTRACK sensor count versus manual mosquito count, in Funchal (coordinates: 32.651318, −16.908259), based on total mosquito captures recorded between 8 June and 23 June 2021.

**Figure 12 biology-14-01047-f012:**
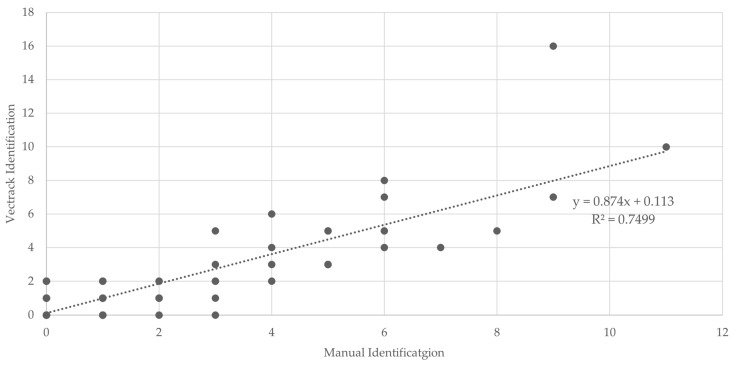
Scatter plot and linear regression analyses comparing VECTRACK sensor identifications with manual identifications, in Funchal (coordinates: 32.651318, −16.908259), considering both genus and sex based on total mosquito captures. Recorded between 8 August 2022 and 28 June 2023.

**Table 1 biology-14-01047-t001:** Total number of mosquitoes captured and identified by manual inspections in Algarve (coordinates: 37.10134887, −8.12356948), by species and sex. The captures occurred between 24 June and 17 September 2021.

Sample Composition		Number of Females	Number of Males	Total
Target species	*Aedes albopictus*	15 (10.5%)	0	15 (10.5%)
	*Culex pipiens*	113 (79.0%)	4 (2.8%)	117 (81.8%)
Non-target species	*Culiseta longiareolata*	8 (5.6%)	2 (1.4%)	10 (7.0%)
	*Anopheles maculipennis*	1 (0.7%)	0	1 (0.7%)
Total Mosquito species		137 (95.8%)	6 (4.2%)	143

**Table 2 biology-14-01047-t002:** Summary of classification metrics on confusion matrices comparing manual identifications with those generated by the VECTRACK system identifications in the Algarve region (coordinates: 37.10134887, −8.12356948). The study took place between 24 June and 17 September 2021.

Classification Task	Number of Samples	Accuracy	Sensitivity	Specificity	Precision
Target and Non-target insects	983	0.971516	0.964	0.967	0.825
Mosquito genus classification	132	0.856	0.964	0.838	0.642
*Aedes* sex classification	15	0.000	0.500	0.000	0.250
*Culex* sex classification	117	0.816	0.885	0.925	0.595

**Table 3 biology-14-01047-t003:** Total number of mosquitoes captured and manually identified in Palmela (coordinates: 38.583248, −8.689730), categorized by species and sex. Sampling was conducted between 8 June 2021 and 23 June 2021.

Sample Composition		Number of Females	Number of Males	Total
Target species	*Aedes caspius*	29 (3.9%)	1 (0.1%)	30 (40.9%)
	*Culex pipiens*	28 (3.8%)	7 (0.9%)	35 (47.8%)
	*Culex theileri*	617 (84.3%)	31(4.2%)	648 (88.5%)
Non-target species	*Culiseta longiareolata*	5 (0.7%)	7 (0.9%)	12 (16.4%)
	*Anopheles maculipennis*	4 (0.5%)	3 (0.4%)	7 (1.0%)
Total Mosquito Species		683 (93.3%)	49 (6.7%)	732

**Table 4 biology-14-01047-t004:** Summary of classification parameters on confusion matrices made for the comparison between manual identifications and VECTRACK system identifications in Palmela (coordinates: 38.583248, −8.689730). Both the captures and the identifications performed by the VECTRACK sensor occurred between 8 June 2021 and 23 June 2021.

Classification Task	Number of Samples	Accuracy	Sensitivity	Specificity	Precision
Target and Non-target insects	1598	0.996	1.000	0.992	0.990
Mosquito genus classification	713	0.967	0.989	0.977	0.742
*Aedes* sex classification	30	0.483	0.983	0.500	0.529
*Culex* sex classification	693	0.885	0.940	0.952	0.690

**Table 5 biology-14-01047-t005:** Total number of mosquitoes captured and identified by manual inspection in Funchal (coordinates: 32.651318, −16.908259) by species and sex. The captures occurred between 8 August 2022 and 28 June 2023.

Sample Composition		Number of Female	Number of Male	Unknown Sex	Total
Target species	*Aedes aegypti*	65 (36.9%)	83 (47.1%)	12 (6.8%)	160 (90.9%)
	*Culex pipiens*	13 (7.4%)	1 (0.6%)	2 (1.1%)	16 (9.1%)
Total		78 (44.3%)	84 (47.7%)	14 (8.0%)	176

**Table 6 biology-14-01047-t006:** Summary of classification metrics on confusion matrices comparing manual identifications with those generated by VECTRACK system identifications in Funchal (coordinates: 32.651318, −16.908259). The study took place between 8 June and 23 June 2021.

Classification Task	Number of Samples	Accuracy	Sensitivity	Specificity	Precision
Target and Non-target insects	367	0.986	0.97	1	0.974
Mosquito genus classification	176	0.928	0.953	0.968	0.758
*Aedes* sex classification	160	0.843	0.851	0.968	0.917
*Culex* sex classification	16	0.687	1	0.266	0.5

## Data Availability

The data supporting the results of this article are included in the paper.

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
