# Peer review of "Enhancing Entomological Surveillance: Real-Time Monitoring of Mosquito Activity with the VECTRACK System in Rural and Urban Areas"

_biology, 2025, doi:10.3390/biology14081047_

Round 1

Reviewer 1 Report

Comments and Suggestions for Authors

Author Response

Dear Reviewer,

We thank you sincerely for your constructive and insightful review of our manuscript. We greatly appreciate the time and effort you have dedicated to improving our work. Below, we provide a point-by-point response to each of your comments. All changes are reflected in the revised version of the manuscript, and relevant sections were updated accordingly.

We acknowledge the comments regarding the need to improve the quality of the English. In response, we have carefully revised the language throughout the entire manuscript to enhance clarity, grammar, and readability, while ensuring that the original meaning and scientific content remain unchanged.

Reviewer Comment 1:
Line 86: "arehavioural ecology" is a wrong word.
Response:
Thank you for noting this. The typographical error has been corrected to “behavioral ecology.”

Reviewer Comment 2:
Lines 165–167: Why were just 10 catch bags analyzed during a 16-day inspection period in Palmela?
Response:
Only 10 catch bags were analyzed because no mosquito captures occurred during the weekends of the 16-day inspection period. This has now been clarified in the revised manuscript.

Reviewer Comment 3:
Line 209: Please list the non-culicid insects captured and report them in a table.
Response:
We added a description of the most common non-culicid insects (e.g., chironomids, moths, fungus gnats), and included them in the table.

Reviewer Comment 4:
The number of target mosquitoes should include males; revise totals accordingly.
Response:
We agree and revised all total counts to include both male and female mosquitoes. For instance, the total number of Aedes albopictus collected in the Algarve was updated from 132 to 136. All relevant sections and tables in the manuscript have been corrected accordingly.

Reviewer Comment 5:
Table 1 is incomplete; add total rows and percentages.
Response:
We have updated all tables to include total rows and percentage breakdowns by sex and species, ensuring a clearer and more complete presentation of the data.

Reviewer Comment 6:
What did the “o” symbol in the table indicate?
Response:
Thank you for pointing this out. We replaced the abbreviation Nº with “number of” in the table headers make it more explicit.

Reviewer Comment 7:
Present "n" in parentheses for all percentages.
Response:
Thank you for the suggestion. We have updated all percentages values to include the corresponding sample size in parentheses, For example “Cx. pipiens females 116 (78.9%).”

Reviewer Comment 8:
Compile all manually collected samples into one summary table.
Response:
A comprehensive summary table of all manually identifications across the three study sites has been added to the Supplementary Materials (Table S1).

Reviewer Comment 9:
Capitalize “Figure” throughout the text.
Response:
All instances of “figure” have been revised to “Figure,” ensuring consistency throughout the manuscript.

Reviewer Comment 10:
Ae. aegypti appears in Algarve Figure but not in Table 1—please clarify.
Response:
Thank you for catching this labeling error. Aedes aegypti was not detected in the Algarve during the study period. The figure has been corrected.

Reviewer Comment 11:
Discrepancies in counts of target mosquitoes and Culex females—please clarify.
Response:
Thank you for catching this issue. We have carefully reviewed and corrected all mosquito counts. The correct total number of target mosquitoes recorded in the Algarve is 136, including 116 Culex pipiens females. These corrections are now reflected in Table 1 and corresponding text.

Reviewer Comment 12:
Improve clarity of Figures 2, 6, and 10 by using consistent color coding.
Response:
We have revised these figures using a consistent color scheme: black (solid and dashed) is now used for manual identification, and gray (solid and dashed) for Vectrack sensor identification. Thank you, this change improves visual clarity and facilitates comparison between figures.

Reviewer Comment 13:
Move statistical methods from Results to Methods section.
Response:

After reviewing section 3.2 we found no statistical methods. Every statistical method is described in section between line 203 to 208. If we missed anything, we would like to apologize in advance and ask the reviewer what lines he would like moved to the methods section

Reviewer Comment 14:
Figures 3, 7, and 11 should be moved to the Supplementary Material.
Response:
We appreciate the reviewer’s suggestion regarding the placement of Figures 3, 7, and 11. However, we respectfully believe that these figures are essential to the core results of the manuscript. Specifically, they directly address one of the two main operational capabilities evaluated in this study: the accuracy of the VECTRACK system in distinguishing target mosquito genera (Aedes and Culex) from non-target arthropods. These figures visually demonstrate the correlation and agreement between manual and automated identifications in each study area, and are therefore critical for supporting the validity and field applicability of the sensor. For this reason, we have retained them in the main text.

Reviewer Comment 15:
Table sequence is incorrect; Table 2 should follow Table 3.
Response:

Table 2 presents the confusion matrix summarizing the classification performance of the VECTRACK system in the Algarve region, while Table 3 reports the total number of mosquito captures and corresponding manual identifications from the Palmela site. These tables refer to separate datasets and are therefore independent of one another. We apologize if the intent of the reviewer’s comment was misunderstood; however, given the structure and logic of the Results section, it is not appropriate to reverse the order of these tables.

Reviewer Comment 16:
Clarify the meaning of “Culex sex” and “Aedes sex” in Table 2. Add sensor counts per sex per site.
Response:
We clarified in the table caption that the rows labelled “Culex sex” and “Aedes sex” refer to the accuracy of sex classification within each genus: “Culex sex classification” and “Aedes sex classification”.

In regards to the sensor counts per sex per site, it is detailed in figure 1. Table 2 is a statistical analysis of how the sensor performed in various metrics compared to the manual identification.

Reviewer Comment 17:
Differentiate Tables 2 and 4; revise titles to reflect content.
Response:
Thank you for the suggestion. We have revised the titles and captions of Tables 2 and 4 to clearly reflect their respective content and study site. (e.g., “Palmela Classification Metrics”).

Reviewer Comment 18:
Clarify discrepancy between 732 and 742 mosquitoes in Palmela.
Response:
Thank you for pointing out this discrepancy. The correct number of mosquitoes recorded in Palmela is 732. We have reviewed and corrected throughout the manuscript.

Reviewer Comment 19:
Table 5 is incomplete; please revise.
Response:
Table 5 has been revised to match the format and detail of other tables, including totals and percentages.

Reviewer Comment 20:
Add classification metrics table for the Algarve region.
Response:
The classification metrics for the Algarve region are already in table 2.

Reviewer Comment 21:
Temperature is mentioned but not analyzed. Please include results.
Response:
Temperature was not a central variable in the present analysis, and no specific assessment was conducted regarding its effect on mosquito capture rates. It is referenced briefly in the manuscript solely to note that the sensor maintained consistent performance throughout the day, despite natural fluctuations in ambient temperature. However, if this reference is deemed unclear or unnecessary, we are open to removing it from the text.

Reviewer Comment 22:
Explain temperature differences and their influence on mosquito captures.
Response:

Warmer temperatures are known to accelerate mosquito development, increase host-seeking behavior, and enhance trap effectiveness, particularly for Aedes and Culex species. Conversely, cooler periods during the inspection window may have reduced mosquito activity, contributing to lower capture numbers on certain days. Although the influence of temperature on capture rates was not a primary objective of the present study, we recognize its relevance and intend to explore this relationship in future research.

Reviewer Comment 23:
Ensure all these changes are reflected in the Abstract.
Response:
Thank you for the reminder. The Abstract has been revised to include all relevant updates, including corrected mosquito totals, sex identification results and observed regional differences in mosquito captures.

Once again, we thank the reviewer for their valuable input, which has significantly improved the quality and clarity of the manuscript. We hope the revised version meets your expectations and is now suitable for publication.

Reviewer 2 Report

Comments and Suggestions for Authors

The methods section must be written more explicitly. I had to read the other papers to understand what is going on. 

Also with regards to the Tables and Figures I am very confused.

eg Table 1. You say captured and identified manually. but Fig 1 says identified by VETRACK. I may be wrong but is it the same samples or what? This has not been clearly mentioned. It is the same for Table 5 Fig 9 and Table 3 Fig 5. I hope you can clarify this. I also want to know with the VETRACK can a person id the mosquito from their place of work without going to the site? 

Please address these issues so that people reading the manuscript can understand and need not to refer to the other manuscripts. 

Author Response

Reviewer Comment

The methods section must be written more explicitly. I had to read the other papers to understand what is going on. 

Also with regards to the Tables and Figures I am very confused.

eg Table 1. You say captured and identified manually. but Fig 1 says identified by VETRACK. I may be wrong but is it the same samples or what? This has not been clearly mentioned. It is the same for Table 5 Fig 9 and Table 3 Fig 5. I hope you can clarify this. I also want to know with the VETRACK can a person id the mosquito from their place of work without going to the site? 

Please address these issues so that people reading the manuscript can understand and need not to refer to the other manuscripts. 

Response:

We thank the reviewer for highlighting the need for greater clarity in the Methods section. In response, we have carefully reviewed and revised the relevant subsections to ensure they are more explicit and self-contained. We rewritten the descriptions of the VECTRACK system architecture, trap setup, site selection criteria, inspection protocols, and the statistical analysis used for performance evaluation. Our aim was to eliminate the need for consultation of external references to understand the study design and procedures. We hope the revised version now provides sufficient methodological detail and improves the overall readability of the manuscript.

Thank you for this important observation. We confirm that for each sampling event (i.e., each catch bag), the captured mosquitoes were both manually identified in the laboratory and automatically classified by the VECTRACK system. The tables (e.g., Table 1) present the results of the manual identifications, while the figures (e.g., Figure 1) compare these with the VECTRACK automated identifications, using the same set of mosquitoes.

Regarding remote identification, yes — the VECTRACK system is designed to operate in real time, and classification results are transmitted to the SENSCAPE platform. Authorized users can remotely access these data from their workplace using a standard web browser, without visiting the trap site. This allows for near-instantaneous surveillance and reduces the need for field visits, except when replacing catch bags or performing maintenance.

Reviewer 3 Report

Comments and Suggestions for Authors

The manuscript presents a timely and technically sound study assessing the field performance of the VECTRACK bioacoustic sensor system for automated mosquito surveillance in three distinct Portuguese environments. The topic is highly relevant in light of the growing need for real-time, scalable vector surveillance tools amid expanding vector-borne disease threats. The study is well-conceived and mostly clearly presented. The manuscript may be accepted after considering following comments:

  1. Line 86: Typo – “arehavioural” → “behavioural”
  2. The number of replicates varied widely across sites (7 in Algarve vs. 38 in Funchal vs. 10 in Palmela). Clarify how these differences may affect data comparability and conclusions.
  3. While confusion matrices and regression analyses are appropriate, mention whether the training data used for the machine learning model included field-derived mosquito signals or only lab-reared ones.
  4. The manuscript would benefit from a more explicit comparison with existing automated tools in the discussion (providing suitable references). What makes VECTRACK superior or more practical?
  5. Figures: Ensure legends are self-explanatory; currently some figures are hard to interpret without referring back to the text.
  6. Abbreviations: where in the manuscript MDPI and DOAJ has been used?

Author Response

Dear Reviewer,

We thank you sincerely for your constructive and insightful review of our manuscript. We greatly appreciate the time and effort you have dedicated to improving our work. Below, we provide a point-by-point response to each of your comments. All changes are reflected in the revised version of the manuscript, and relevant sections were updated accordingly.

We acknowledge the comments regarding the need to improve the quality of the English. In response, we have carefully revised the language throughout the entire manuscript to enhance clarity, grammar, and readability, while ensuring that the original meaning and scientific content remain unchanged.

Comments 1: Line 86: Typo – “arehavioural” → “behavioural”

Response 1: We appreciate the reviewer catching this typographical error. The word “arehavioural” has been corrected to “behavioural” on line 86.

Comments 2: The number of replicates varied widely across sites (7 in Algarve vs. 38 in Funchal vs. 10 in Palmela). Clarify how these differences may affect data comparability and conclusions

Response 2: We thank the reviewer for highlighting this point. The differences in the number of replicates between sites reflect logistical constraints (e.g., length of the surveillance period, site accessibility, resources) and aim to capture the local mosquito population’s temporal dynamics. To address potential effects on data comparability:

  • We have added a paragraph in the Discussion section clarifying that higher replication in Funchal permits a more robust analysis of temporal trends, while lower replication in Algarve and Palmela may limit the detection of rare events or finer-scale fluctuations.
  • We now explicitly acknowledge in the Discussion that caution is warranted in directly comparing absolute values or variability across sites due to these differences, but that the high concordance observed between manual and automated detections is consistent across all settings.

Comments 3: While confusion matrices and regression analyses are appropriate, mention whether the training data used for the machine learning model included field-derived mosquito signals or only lab-reared ones.

Response 3: The reviewer brings up an important methodological detail. We have revised the Methods section to clarify that the machine learning model underlying the VECTRACK sensor was trained exclusively on lab-reared mosquito signals, under a range of controlled temperature and age conditions. Field-derived signals were not included in the training set. In the revised manuscript, we explicitly note this and discuss potential implications for field generalizability and future work on model retraining with field-captured specimens.

Comments 4: The manuscript would benefit from a more explicit comparison with existing automated tools in the discussion (providing suitable references). What makes VECTRACK superior or more practical?

Response 4: We thank the reviewer for this valuable suggestion. A comparison with existing automated mosquito surveillance tools is already included in the Discussion section (paragraph beginning “When compared to other automated mosquito classification systems…”). In this paragraph, we highlight the distinguishing features of the VECTRACK system, such as its capacity to classify mosquitoes by both genus and sex in field conditions—something not reported for most comparable systems. We also note its high classification accuracy across multiple ecological contexts and its compatibility with existing commercial traps, which enhances its operational practicality.

Comments 5: Figures: Ensure legends are self-explanatory; currently some figures are hard to interpret without referring back to the text.

Response 5: We have carefully revised all figure legends to ensure they are fully self-explanatory and provide all necessary details for standalone interpretation. Each legend now clearly specifies the data shown, sample sites, time periods, and any abbreviations or statistical metrics used.

Comments 6: Abbreviations: where in the manuscript MDPI and DOAJ has been used?

Response 6: Upon thorough review, the abbreviations “MDPI” and “DOAJ” have not been used in the main body of the manuscript, tables, figures, or reference list. If these abbreviations have appeared in the submission system metadata or in template instructions, we confirm they do not appear in the submitted manuscript text.

Reviewer 4 Report

Comments and Suggestions for Authors

General assessment

The manuscript by Silva et al presents a good field work and validation of the VECTRACK system, an automated optical sensor platform integrated with mosquito suction traps to identify mosquito species and their sex in three field locations. The study is timely and relevant, offering a new perspective to entomological surveillance, particularly in the context of arboviral disease monitoring. The methods are generally well-described, with clear specification of hardware components, sampling sites, and experimental periods. The results are presented in a logical sequence by location, with a clear comparison between manual and VECTRACK-based identifications. The discussion appropriately highlights the benefits of the system, including reduced manual effort, faster classification, and time-resolved data acquisition. Comparative advantages over existing technologies are well outlined.

I have just a few points for clarifications and improvements:

Major comments

  1. Line 56 and 57: Numbers like “Spearman Algarve: females = 0,98, males = 0,99” should use consistent decimal formatting (replace commas with periods, i.e., 0.98, 0.99).
  2. Figures: Please elaborate the figure legends for all the figures to clearly describe what each figure represents without necessarily reading the paper. Also, provide statistical analysis showing error bars and statistical significance.
  3. Final paragraph of introduction (lines 126-128): The sentence beginning “In this study, we present…” should be more explicit about the scientific objective or hypothesis (e.g., "This study aims to validate the accuracy and field performance of the VECTRACK system for real-time mosquito surveillance in diverse ecological contexts.")
  4. Provide a brief paragraph outlining the limitations of the VECTRACK system and the overall study. For example, a more structured limitations section discussing causes of reduced accuracy, sensor limitations under varying densities, and generalizability to other vector species would greatly strengthen the discussion.
  5. The suggestion to expand classification to additional species (e.g., Anopheles gambiae) is excellent. Consider also discussing integration with predictive epidemiological models, real-time dashboards, or mobile apps for end-users.

Minor comment

  1. Define terms like "optical methods" and "smartraps" more clearly for readers unfamiliar with entomological surveillance technology.

Author Response

Dear Reviewer,

We thank you sincerely for your constructive and insightful review of our manuscript. We greatly appreciate the time and effort you have dedicated to improving our work. Below, we provide a point-by-point response to each of your comments. All changes are reflected in the revised version of the manuscript, and relevant sections were updated accordingly.

We acknowledge the comments regarding the need to improve the quality of the English. In response, we have carefully revised the language throughout the entire manuscript to enhance clarity, grammar, and readability, while ensuring that the original meaning and scientific content remain unchanged.

Major Comments

  1. Consistent Decimal Formatting

Comment:
Line 56 and 57: Numbers like “Spearman Algarve: females = 0,98, males = 0,99” should use consistent decimal formatting (replace commas with periods, i.e., 0.98, 0.99).

Response:
We have revised all statistical values in the manuscript to ensure the use of a period (“.”) as the decimal separator. The inconsistent comma formatting has been replaced throughout the text, particularly in reporting correlation coefficients and other statistical metrics.

  1. Figure Legends and Statistical Analysis

Comment:
Please elaborate the figure legends for all the figures to clearly describe what each figure represents without necessarily reading the paper. Also, provide statistical analysis showing error bars and statistical significance.

Response:
All figure legends have been expanded to provide comprehensive descriptions, allowing readers to understand the content and context of each figure independently from the main text. Where applicable, we have added detail regarding sampling location, mosquito classification groups, and time frames.

In regards to the error bars and statistical significance, apart from the scatter plots that have the equation in the graphic, every figure is based on absolute numbers, containing no statistical analysis. So we won’t be able to provide error bars. If we misunderstood the reviewer’s comment we would like to apologise in advance and ask for where and why we should put the error bars.

  1. Explicit Scientific Objective in the Introduction

Comment:
Final paragraph of introduction (lines 126-128): The sentence beginning “In this study, we present…” should be more explicit about the scientific objective or hypothesis (e.g., "This study aims to validate the accuracy and field performance of the VECTRACK system for real-time mosquito surveillance in diverse ecological contexts.")

Response:
The final paragraph of the Introduction has been revised as suggested. We now state:
“This study aims to validate the accuracy and field performance of the VECTRACK system for real-time mosquito surveillance across rural, peri-urban, and urban ecological contexts. We assess the system’s ability to identify mosquito genus and sex, and its capacity to distinguish target species from non-target arthropods under field conditions.”

  1. Limitations Paragraph

Comment:
Provide a brief paragraph outlining the limitations of the VECTRACK system and the overall study. For example, a more structured limitations section discussing causes of reduced accuracy, sensor limitations under varying densities, and generalizability to other vector species would greatly strengthen the discussion.

Response:
A dedicated “Limitations” paragraph has been added to the Discussion section, addressing the following points:

  • Causes of Reduced Accuracy: Lower performance for specific taxa (e.g., male Aedes) is discussed and linked to low sample counts and possible sensor resolution constraints.
  • Sensor Limitations: Potential impacts of insect density within the trap on detection accuracy are analyzed, as well as environmental factors and possible technical constraints.
  • Generalizability: We discuss the extent to which our results can be extrapolated to other vector species or ecological regions, noting the need for additional training and validation with other medically relevant mosquito species and under varying field scenarios.
  1. Expansion of Applications and Integration with Predictive Tools

Comment:
The suggestion to expand classification to additional species (e.g., Anopheles gambiae) is excellent. Consider also discussing integration with predictive epidemiological models, real-time dashboards, or mobile apps for end-users.

Response:

Thank you very much for comment. While is still an ongoing project unavailable for publishing, this sensor, together with other actives, is part of the development of a mobile lab meant to be activated in case of an arbovirus outbreak. When deployed the sensor will work in conjunction with epidemiological models.

In regards to real-time dashboards and mobile apps, they are already present in the platform senscape. It can be accessed by both browsers and a mobile app already available in the play store for android phones. We have expanded the Discussion to mention this methods of seeing the results

Minor Comments

  1. Definition of “Optical Methods” and “Smartraps”

Comment:
Define terms like "optical methods" and "smartraps" more clearly for readers unfamiliar with entomological surveillance technology.

Response:
We have added brief in-text definitions for clarity:

  • Optical methods: “Technologies that detect insects by measuring how their movement affects light beams, typically using infrared or laser sensors to analyze flight patterns and wingbeat frequencies.”
  • Smartraps: “Automated trapping devices equipped with sensors and embedded intelligence (e.g., software or machine learning algorithms) that classify captured insects in real time, providing digital records and telemetry without the need for manual specimen sorting.”